# High Strain Rates Impact Performance of Glass Fiber-Reinforced Polymer Impregnated with Shear-Thickening Fluid

Minghai Wei [1], Li Sun [2],*, and Wanjin Gu [2]

1 School of Civil and Architecture, Zhejiang Sci-Tech University, Hangzhou 310018, China; wei.mgh@zstu.edu.cn
2 School of Civil Engineering, Shenyang Jianzhu University, Shenyang 110180, China
* Correspondence: sunli2009@163.com

**Abstract:** This paper examines the behavior at high strain rates of a shear-thickening fluid (STF) impregnated glass fiber-reinforced polymer (GFRP) fabric using a split Hopkinson pressure bar (SHPB). This study involved impact testing of 4 GFRP specimens and 20 GFRP-STF composite specimens at four different strain rates. The STF employed in this study was synthesized by incorporating 20.0 wt.% of 12 nm silica in polyethylene glycol. Rheological tests indicated that the STF exhibited a noticeable shear-thickening effect, with viscosity surging from 3.0 Pa·s to 79.9 Pa·s. The GFRP-STF specimen demonstrated greater energy absorption capacity, deformation ability, and toughness, bearing higher and faster impact loads than neat GFRP. Specifically, the GFRP-STF specimen showed a 21.8% increase in peak stress and a 92.9% rise in energy absorption capacity under high-strain-rate loading. Notably, the stress–strain curve of the GFRP-STF specimen exhibited a distinct yield stage, while the energy absorption curve displayed no significant descending stage features.

**Keywords:** shear-thickening fluid; GFRP; impact performance; high strain rate; SHPB

## 1. Introduction

Glass fiber-reinforced polymer (GFRP) composites have been widely used in strengthening and reinforcing structural components of existing civil engineering, owing to their excellent physical and mechanical properties under quasistatic loads [1,2]. However, GFRP is prone to brittle fractures under impact loads due to transverse cracking in its matrix or debonding of the fiber/matrix interface, which leads to premature failure. [3,4]. To improve the dynamic performance of a GFRP under impact loading, studies have developed GFRP impregnated with shear-thickening fluid (GFRP–STF) composite materials and their mechanical properties have been examined under high- and low-velocity impacts [5–12]. Nevertheless, under extreme impact loads caused by explosions, vehicular and ship collisions, rockfall impacts, and other reasons, the ultimate failure strength and energy absorption capacity of a GFRP–STF composite may vary considerably owing to the strain-rate effect. Therefore, it is very important to study the mechanical properties of GFRP–STF composites under high-strain-rate load impacts for aiding the structural design of these materials.

A STF is a new nanosuspension formed by uniformly dispersing micro–nano particles in a dispersant [13,14]. It is easy to deform under low-shear-rate loads, but under high-velocity external impacts, such as high-shear-rate impacts, it rapidly transforms into a firm solid. Following the removal of the impact force, STF reverts from its solid phase to its original liquid state [15,16]. Thus, a STF absorbs a large amount of impact energy, and can be extensively used in soft armor composite materials [17–21], vibration control [22–27], and other fields [28–31].

Based on these properties of STFs, Hasan-Nezhad et al. [5] investigated the mechanical properties of low-velocity impact resistance of GFRP–STF composites based on yarn pulling out, quasistatic puncture, flexibility, and thickness tests. Their research showed that when the flexibility of a GFRP–STF fabric was insignificantly changed compared to that of the neat GFRP, the yarn drawing and puncture properties of the former increased by 353% and 45%, respectively. Consequently, the shear impact resistance increased by 130%. Hasan-Nezhad et al. [6] subsequently investigated the ballistic and cushioning properties of a GFRP impregnated with a pure STF and a treated STF (TSTF) based on air gun and drop hammer impact tests, respectively. It was observed that five layers of the GFRP fabric impregnated with 50 wt% TSTF yielded the best cushioning performance, and its peak stress was reduced to 60% in drop hammer impact tests. Simultaneously, the ballistic tests showed that the STF effectively reduced the penetration depth of the GFRP, whereas the rheological performance of the STF had a considerable impact on the ballistic performance. Regarding the compression responses of GFRP–STF composites, Jeddi and Yazdani [7] found that a STF improves the compression properties of a GFRP, which are enhanced with the increase in the STF concentration and the modification of the rheological properties. On this basis, Jeddi and Yazdani [8] reported that the STF concentration also had a considerable effect on the puncture resistance of the GFRP–STF fiber fabric. To further improve the quasistatic puncture resistance of a GFRP–STF composite, Balali et al. [9] conducted comparison tests of this property between GFRP–STF and GFRP impregnated with nanoclay/STF (GFRP–clay/STF) composites. It was found that the addition of the nanoclay in the STF considerably improved the penetration resistance of the GFRP–STF fabric. For example, when the content of the nanoclay was 3%, the maximum resistance and energy absorption capacity of the GFRP–clay/STF composite were increased by 29.8% and 31.3%, respectively.

The effects of physical parameters of a GFRP on its impact mechanical properties were not considered in the above studies. Wei and Sun et al. [10] studied in detail the effects of laying layers, angles, and constraints of GFRP–STF composites on their low-velocity impact resistance mechanical properties based on drop hammer impact tests. The results revealed that when the number of layers was four or the laying angle was 45°, the maximum resistance and energy absorption of the GFRP–STF composites were significantly improved. It was also found that almost all GFRP–STF specimens failed owing to the fiber fracture being different from that of the neat GFRP [11]. Additional studies confirmed that compared with carbon fiber-reinforced polymer composites, a GFRP impregnated with a STF has significant energy absorption capacity. Moreover, it shows a change in the mechanism of GFRP resistance to low-velocity impact mechanics, which is caused by the filling and adherence of the dispersed medium in the STF. In the aforementioned studies, the preparation of a GFRP–STF composite was modeled based on Kevlar–STF composite materials. The impact resistance of a GFRP–STF composite can only be influenced by the GFRP or the STF. Selver [12] developed a new method to improve the impact resistance of a GFRP–STF composite. When impregnated with a STF, a GFRP was transformed into a thermosetting composite by vacuum injection of an epoxy resin. Experimental studies confirmed that the incorporation of STF not only resulted in a shift to the interface bonding and failure mechanism of GFRP but also heightened the residual strength and damage tolerance of GFRP–STF composites relative to pure GFRP.

Previous studies on GFRP-STF composite materials have explored their low-impact properties via drop hammer, puncture, and pull-out tests. However, these studies did not investigate the material's high-strain-rate response and neglected to consider the influence of GFRP physical characteristics on their high-strain-rate impact mechanical properties. The objective of this study was to investigate the high-strain-rate behavior of GFRP-STF composites, using split Hopkinson pressure bar (SHPB) tests on four neat GFRP and twenty GFRP-STF specimens at varying air pressures. Moreover, this study examines the effects of the number of layers and laying angle on the stress–strain and energy-absorption responses of the GFRP-STF specimens.

## 2. Experiments and Methods

### 2.1. Materials

Figure 1a shows the scanning electron microscopy (SEM) image of the nanosilica particles used in this study. The primary particle size, relative density, and pH range of the nanosilica were 12 nm, 2.319–2.653, and 3.7–4.7, respectively. Polyethylene glycol (PEG200) is a stable transparent liquid at room temperature, and its hydroxyl value is 510–623 mgKOH/g. The dispersant was N-aminoethyl-γ-aminopropyltrimethoxysilane, with a molecular weight of 222.36, which was purchased from the Tianjin Guangfu Fine Chemical Research Institute.

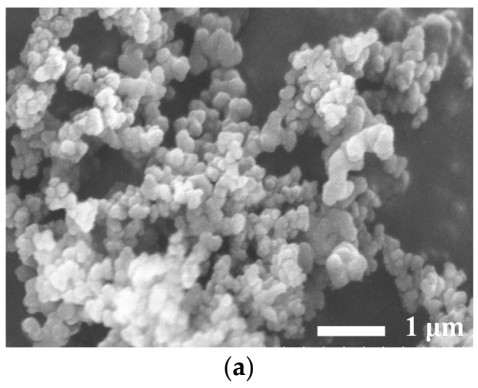

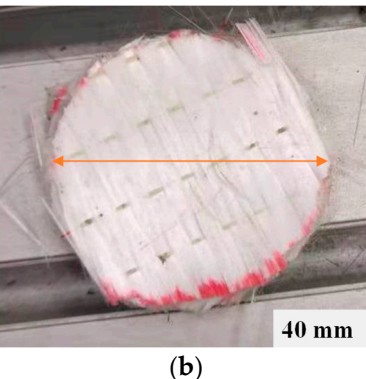

(**a**)  (**b**)

**Figure 1.** Images of $SiO_2$ nanoparticles and GFRP specimen. (**a**) SEM of $SiO_2$ nanoparticles; (**b**) GFRP specimen.

The GFRP (E-type) used in all impact tests was produced by the Tongshen Corporation [see Figure 1b]. The diameter of the GFRP for impregnation with a STF was 40 mm. The aerial density, monolayer thickness, and limited elongation were 430 $g/m^2$, 0.17 mm, and 2.8%, respectively.

### 2.2. STF Preparation and Rheological Response

The STF was prepared by dispersing the purchased high-quality silica nanoparticles in PEG200 by ultrasonication and mechanical agitation. During this process, the ultrasonic oscillator (Guanbo Technology, Shenzhen, China) was used continuously until the dispersed phase particles were uniformly dispersed in PEG200. The configured STF was placed in a vacuum drying oven (Lichen, Shanghai, China) at 110 °C, and a stable 20 wt% STF system was obtained after the bubbles in the system were removed.

The rheological properties of the 20 wt% STF were tested by an AR2000 rheometer (TA Instruments, Newcastle, DE, USA). The specific tests were as follows: room temperature 25 °C, plate rotor diameter 25 mm, plate spacing 0.1 mm, and shear rate scanning range 0.01–1000 $s^{-1}$. Figure 2 shows the variation in the viscosity of the STF as a function of the shear rate (from 1.0 $s^{-1}$ to 1000 $s^{-1}$). It can be observed that when the shear rate is low, the viscosity exhibits shear thin characteristics; however, when the shear rate increases to a certain critical value, the viscosity increases abruptly. The peak viscosity increases to 79.9 Pa·s, and the shear thickening effect is 28.5. Further, as evident from the shear stress–shear rate curve, the stress of the STF is also a function of the shear rate. When plotted on the logarithmic scale, the shear stress rapidly increases with the increase in the shear rate. In addition, when the shear rate increases to 79.4 $s^{-1}$, the shear stress also exhibits a remarkable abrupt transition, and the shear stress increases 65.4 times relative to that at the takeoff point. The findings presented above show that the STF prepared in this study induces a significant shear-thickening effect.

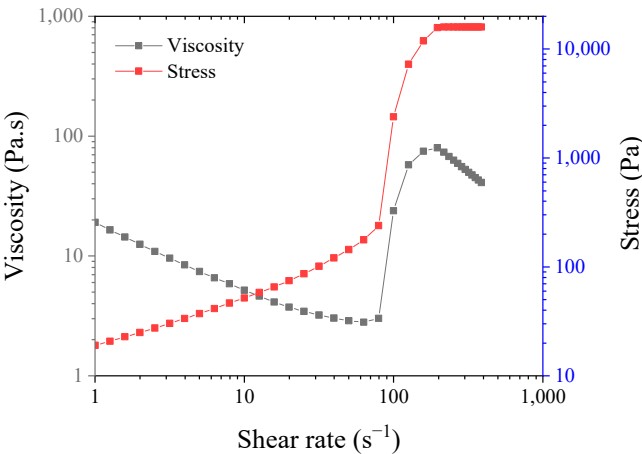

**Figure 2.** Rheological results of STF.

### 2.3. Impregnation of GFRP and Microscopy Analysis

The impregnation of the GFRP for the preparation of the GFRP–STF composite material was facilitated by diluting the STF in absolute ethanol at a 1:1 vol ratio and soaking individual GFRP layers in it for 10 min. After impregnation with the absolute ethanol/STF mixture, the GFRP layers were heated in an oven at 85 °C for 30 min to remove the absolute ethanol.

Surface morphologies of neat GFRP and GFRP–STF specimens at different magnifications are shown in the SEM images in Figure 3. As shown in Figure 3a,b, the filaments of the GFRP are smooth, and there are some gaps among them. However, referring to Figure 3c,d, the filaments of the GFRP–STF specimen are covered with numerous nanoparticles. Moreover, the gaps between the filaments of the GFRP–STF specimen are filled by nanoparticles in a significantly uniform manner.

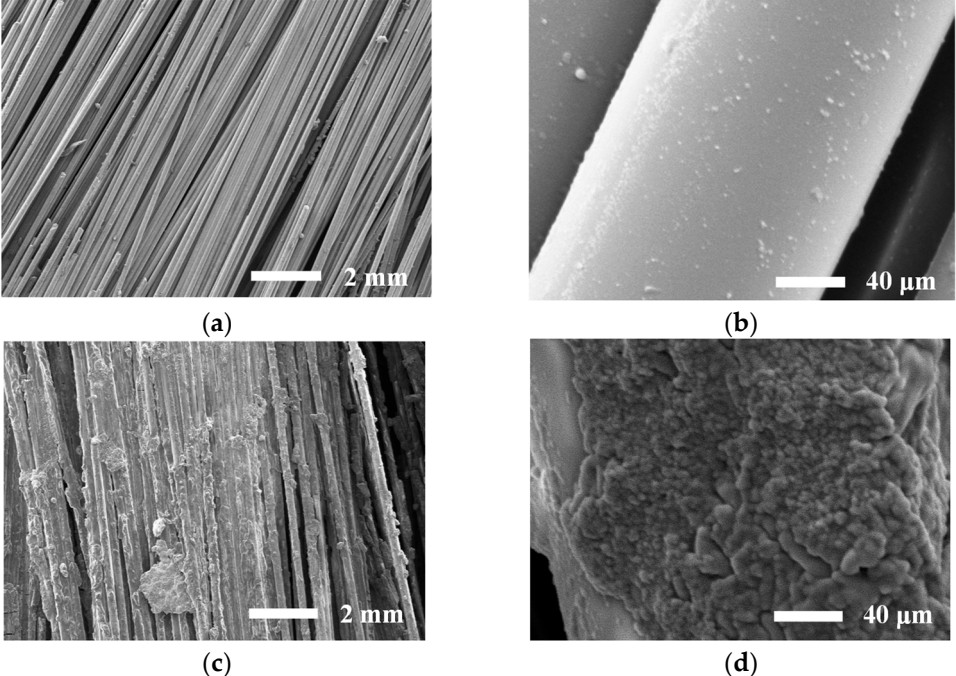

**Figure 3.** SEM images of GFRP and GFRP-STF specimens. (**a**) Neat GFRP-500 μm; (**b**) neat GFRP-10 μm; (**c**) GFRP-STF-500 μm; and (**d**) GFRP-STF-10 μm.

### 2.4. SHPB Impact Test Setup and Procedures

The Hopkinson bar test system [32] used in this study is shown in Figure 4. The entire test system consists of three parts: a loading device, pressure bar device, and data acquisition system. The loading device uses high-pressure pure nitrogen as the power source. To ensure the repeatability of the test, the loading depth of the bullets was kept unchanged in the test. Only the pressure in the air gun was adjusted by the pressure-reducing valve of the nitrogen cylinder. The strain rate corresponding to the air pressure was subsequently calculated by the data acquisition system. The strain rates were 3800 s$^{-1}$, 4100 s$^{-1}$, 5100 s$^{-1}$, and 6100 s$^{-1}$.

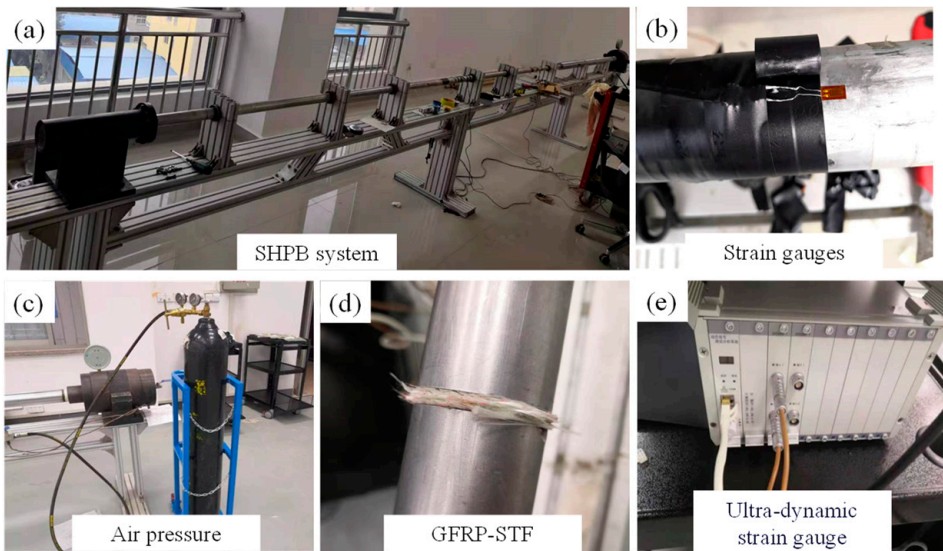

**Figure 4.** SHPB impact test setup.

The pressure bar device consisted of an incident bar, a transmission bar, an absorption bar, and an end-damping device. In the tests, aluminum rods were used (diameter 40 mm, elastic modulus 70 GPa, density 2.71 g/cm$^3$), and the lengths of the incident rods and transmission shaft were 1800 mm. Considering the use of a shaper to control the loading wave, the impact rod length was 300 mm. In addition, a cylindrical red copper pulse shaper with a thickness of 1 mm and a diameter of 40 mm is added in front of the incident rod, and the stress and strain of the test piece are measured with a dynamic strain gauge (DHDAS, V8.0, Donghua Software, China) and a strain gauge. In order to ensure the accuracy of the output voltage in the test, a 120 $\Omega$ strain gauge is used in the test, and the bridge voltage is 2 V. The strain gauge is shown in Figure 4b, and the dynamic strain gauge is shown in Figure 4e.

The strain on the incident bar ($\varepsilon_R$) and the transmission bar ($\varepsilon_T$) are collected using a strain gauge. The strain, strain rate, and stress of the specimen can be obtained from Equation (1a,b). Then, the stress–strain curve is obtained from Equation (2) [33,34].

$$\varepsilon(t) = -\frac{2C_0}{L} \int_0^t \varepsilon_R(t)\,dt \tag{1a}$$

$$\dot{\varepsilon}(t) = -\frac{2C_0}{L}\varepsilon_R(t) \tag{1b}$$

$$\sigma(t) = E\frac{A}{A_0}\varepsilon_T(t) \tag{2}$$

where $\sigma$ and $\varepsilon$, respectively, refer to the stress and strain of the specimen; $C_0$, $A$, and $E$ are the wave velocity, cross-sectional area, and elastic modulus of the elastic compression bar, respectively; and $L$ and $A_0$ are the original length and cross-sectional area of the sample.

In addition, the energy absorption capacity of the CFRP-STF was calculated using Equation (3) based on the stress–strain curve obtained from the experimental results.

$$W = \int_{0}^{\varepsilon} \sigma(\varepsilon)d\varepsilon \tag{3}$$

where $W$ is the amount of energy absorption.

## 3. Results and Discussion

### 3.1. Impact Response of GFRP–STF Specimen under High Strain Rate

Figure 5 shows the stress–strain responses of the neat GFRP and GFRP–STF specimens at different strain rates. Noticeably, the stress–strain curves of both show the same response behaviors with increasing strain rate, and both exhibit the "uplift" phenomenon, indicating that both are associated with significant strain-rate effects. However, comparing Figure 5a,b, it can be inferred that the peak stress of the GFRP–STF composite is higher than that of the neat GFRP at all strain rates except at the strain rate of 3800 s$^{-1}$. At this strain rate, it is lower than that of the neat GFRP. The increased ranges are 21.8%, 14.9%, and 6.1%, respectively, indicating that the GFRP impregnated with the STF can withstand higher and faster impact loads than the neat GFRP.

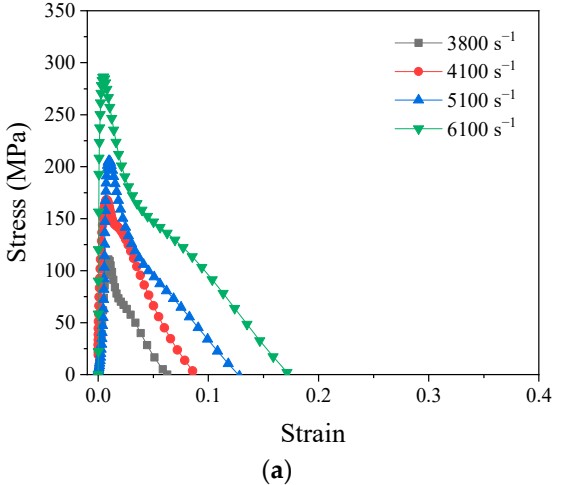
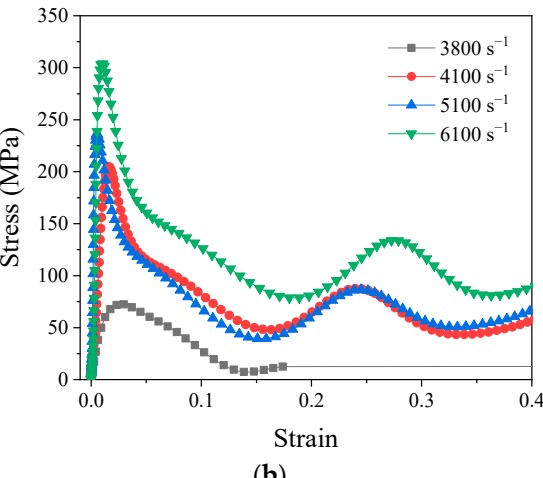

**Figure 5.** Stress–strain response of neat GFRP and GFRP-STF specimens under different strain rates (all test specimens have three layers and 90° laying). (**a**) Neat GFRP; (**b**) GFRP-STF.

More importantly, the immersion of the STF significantly changed the impact mechanical behavior of the GFRP. From the comparison between Figure 5a,b, it can be observed that the stress–strain curve of the GFRP–STF composite does not fail at small strains, whereas it presents a notable yield stage, particularly when the strain rate is high. This indicates that the GFRP–STF composite has good toughness. As shown in Figure 3, this can be attributed to the GFRP impregnated with the STF containing numerous nanoparticles filling and covering the fibers. The existence of these nanoparticles increases the friction between the GFRP fibers, allowing the fibers to jointly bear the impact load, and hinders the transmission of the impact load among GFRP fibers.

Figure 6a,b show the curves of energy absorption with time of neat GFRP and GFRP–STF specimens under four high strain rates, respectively. In these figures, the energy absorption capacities of the neat GFRP and GFRP–STF specimens increase with the increasing strain rate, indicating a significant strain-rate effect. Further, comparing Figure 6a,b, it

can be observed that consistent with the stress–strain response in Figure 5, the peak energy absorption of the GFRP–STF specimen is slightly reduced at the strain rate of 3800 s$^{-1}$. In contrast, those at the other strain rates are higher than those of the neat GFRP, with the most considerable increase occurring at the strain rate of 4100 s$^{-1}$ by approximately 92.9%. This may be attributed to the fact that at this strain rate, as shown in Figure 7b, the neat GFRP is close to the failure state, in which the GFRP fibers are largely broken and cannot absorb the impact energy by densification. However, in the GFRP–STF specimen, because numerous nanoparticles in the STF fill or adhere to the fibers, the fiber fragments can be densified again by the friction and adhesion among the nanoparticles when the GFRP fibers are largely broken. Consequently, more impact energy can be absorbed (there is no remarkable decrease in the peak value, as shown in Figure 6b). Therefore, the STF induces a significant energy absorption enhancement effect at this strain rate. When the strain rate increases to 5100 s$^{-1}$ and 6100 s$^{-1}$, as shown in Figure 7c,d, the GFRP–STF composite breaks in a large area, indicating that the impact energy at this strain rate is much greater than that withstood by the GFRP–STF composite. Therefore, the addition of STF has only a minor effect on increasing the energy absorption capacity of GFRP, with an increase of approximately 10%.

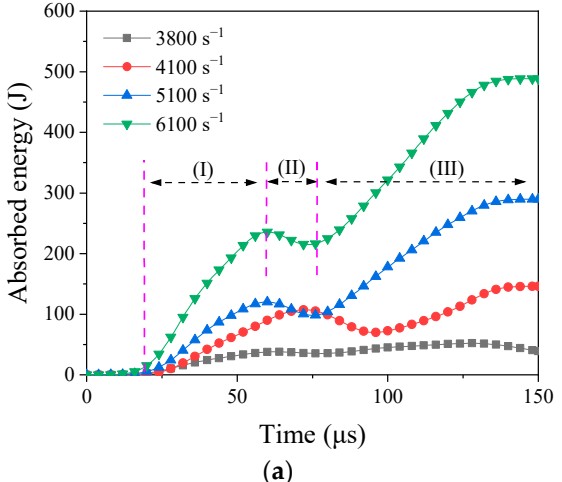

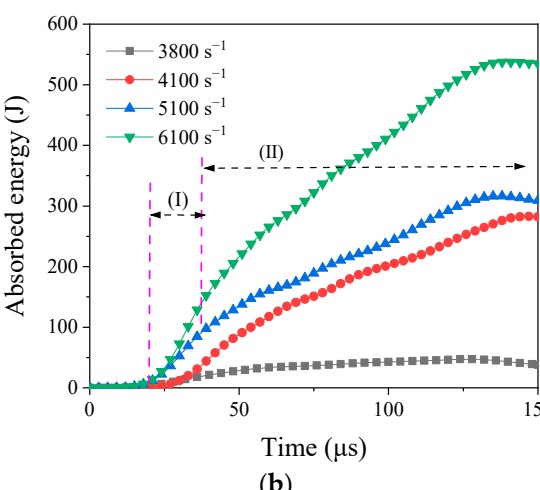

(**a**)  (**b**)

**Figure 6.** Absorption energy–time of neat GFRP and GFRP-STF specimens under different strain rates (all test specimens have three layers and 90° laying). (**a**) Neat GFRP; (**b**) GFRP-STF.

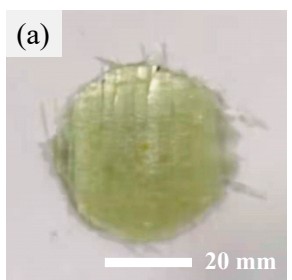
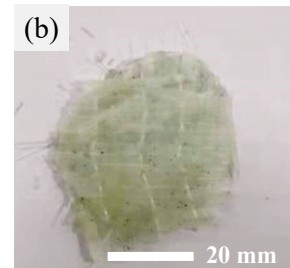
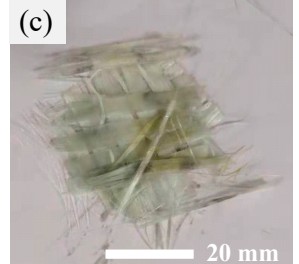
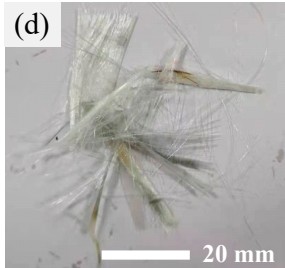

**Figure 7.** Macroscopic failure of GFRP-STF specimens under different strain rates: (**a**) 3800 s$^{-1}$, (**b**) 4100 s$^{-1}$, (**c**) 5100 s$^{-1}$, and (**d**) 6100 s$^{-1}$ (all test specimens have three layers and 90° laying).

The STF not only improves the energy absorption capacity of the GFRP but also changes its energy absorption behavior. Taking the strain rate of 6100 s$^{-1}$ as an example, it can be observed from Figure 6a that the energy absorption curve of the GFRP is mainly divided into three sections, and the curve in the first (I) stage increases linearly toward the apex, which corresponds to the elastic behavior of the structure [35]. The curve in the second (II) stage shows that the energy absorption capacity decreases owing to fiber rupture. The curve in the third (III) stage presents that the energy absorption capacity is further significantly increased owing to the densification caused by fiber crushing until stabilization [36]. However, as shown in Figure 6b, the energy absorption curve of the GFRP–STF composite is mainly divided into two sections, and the curve in the first (I) stage increases linearly until the inflection point of the curve. However, in the second (II) stage, owing to the existence of numerous nanoparticles in the GFRP–STF specimen, its energy absorption curve no longer contains a notable downward section, and it increases significantly approximately linearly until it is stabilized.

Figure 7 shows the field photographs of GFRP–STF specimens after impact tests under different high-strain-rate loadings. As shown in Figure 7a, there are impact marks and only a few burrs (fiber fracture) on the GFRP–STF surface; however, the damage is unremarkable. When the strain rate increases to 4100 s$^{-1}$, as shown in Figure 7b, the GFRP–STF specimen presents notable breaking characteristics. First, the specimen surface appears silver white owing to the increased number of burrs, and second, a large area exhibits disintegration around the sample. Comparatively, Figure 7b,c show that the GFRP–STF composite is damaged under loading at a strain rate of 5100 s$^{-1}$. Specifically, the entire sample is completely disintegrated except near the central point, and there are many burrs in the disintegrated parts. The most important finding is that the volume of the GFRP–STF composite is significantly smaller than of the specimens in Figure 7a,b. Figure 7d shows the photograph of the GFRP–STF composite under loading at the strain rate of 6100 s$^{-1}$. It can be observed that it is completely broken, and only a few filaments remain.

### 3.2. Influence of Laying Angle on the Strain-Rate Effect of GFRP–STF Specimen

Table 1 and Figure 8 show the effect of the laying angle on the strain-rate effect of the GFRP–STF composite. As noticeable from Figure 8a,b, when the laying angles of GFRP–STF specimens are 45° and 90°, the peak stress and the peak energy absorption are increased compared to those when the laying angle is 0°. Moreover, this increased ability is related to the strain rate. Based on Figure 8a, 45° has a more significant advantageous effect than 90° in increasing the peak stress of the GFRP–STF composite, and this advantage is more notable before it breaks. For example, compared with the 0°-laying GFRP–STF specimen, the peak stress of the 45°-laying composite is increased by 130.2% and 81.3% when the strain rates are 3800 s$^{-1}$ and 4100 s$^{-1}$, respectively. These are 2.4 and 1.3 times higher than those of the 90°-laying composite, respectively. However, it can be observed from Figure 8b that the GFRP–STF specimen laid at 90° has the best advantage in energy absorption capacity, which is unremarkable compared to the GFRP–STF laid at 45°. For example, when the strain rate is 6100 s$^{-1}$, its improvement rate is only 6.0%.

**Table 1.** Effect of laying angle on the high strain rate impact behavior of GFRP-STF composites.

| Strain Rates (s$^{-1}$) | Peak Stress (MPa) | | | Peak Absorption Energy (J) | | |
|---|---|---|---|---|---|---|
| | 20%-3-0° | 20%-3-45° | 20%-3-45° | 20%-3-0° | 20%-3-45° | 20%-3-90° |
| 3800 | 46.9 | 108 | 72.4 | 52.2 | 76.6 | 47.3 |
| 4100 | 125.3 | 227.2 | 204.8 | 161.6 | 272.8 | 282.8 |
| 5100 | 237.6 | 255.5 | 236.5 | 239.4 | 352.9 | 376.4 |
| 6100 | 267.3 | 322 | 303.9 | 472.7 | 507.4 | 537.7 |

Note: the number 20% denotes the mass fraction of STF; the number 3 represents the number of layers; and the numbers 0°, 45°, and 90° all represent the layer angle.

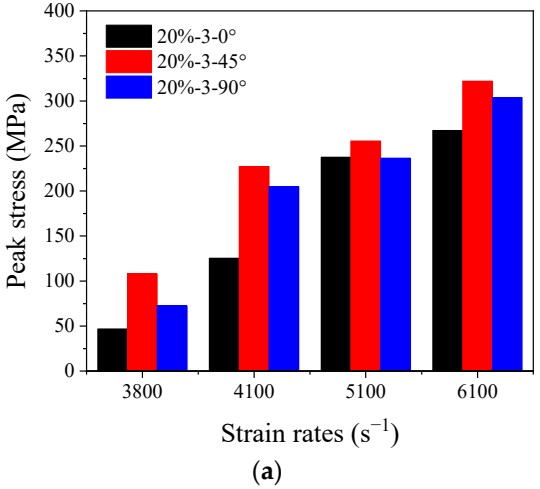
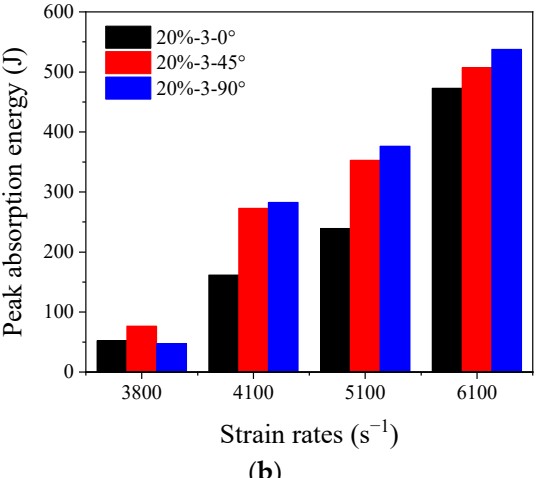

**Figure 8.** Effect of the laying angle on the impact response for GFRP-STF specimen under different strain rates. (**a**) Peak stress; (**b**) peak absorption energy.

*3.3. Influence of Layer Number on the Strain-Rate Effect of GFRP–STF Specimen*

Table 2 and Figure 9 show the effect of the number of layers of the GFRP–STF composite on its peak stress and energy absorption capacity at different strain rates. Different from the effect of the laying angle, the effect of the laying layers has notable regularity. For example, both the peak stress and energy absorption capacity of a GFRP–STF increase with increasing layers. However, it can be observed from Figure 9a that the number of layers has little effect on the increase in the peak stress, particularly at high strain rates ($6100 \text{ s}^{-1}$). This can be attributed to the fibers of three, four, and five layers of the GFRP–STF composite being broken and losing their bearing capacities under the loading at this strain rate ($6100 \text{ s}^{-1}$). Nevertheless, owing to the impregnation with the STF, the GFRP realizes a better deformation capacity. Therefore, the increase in the number of layers significantly improves the energy absorption capacity of the GFRP at all strain rates, particularly at low strain rates. As shown in Figure 9b, the energy absorption capacity of the GFRP–STF composite with four layers increased by 482.7% compared to that of the GFRP–STF specimen with three layers. However, the energy absorption capacity of the five-layer GFRP–STF specimen is only 9.3% higher than that of the four-layer GFRP–STF, which is mainly attributed to the insufficient impact energy at this strain rate. When the strain rate is further increased, the improved energy absorption capacity of the five-layer GFRP–STF composite relative to that of the four-layer one is fully reflected, as shown in Figure 9b. Moreover, at the strain rates of $4100 \text{ s}^{-1}$, $5100 \text{ s}^{-1}$, and $6100 \text{ s}^{-1}$, its energy absorption capacity is increased by 50.0%, 39.9%, and 27.6%, respectively.

**Table 2.** Effect of the number of layers on the high strain rate impact behavior of GFRP-STF composites.

| Strain Rates ($s^{-1}$) | Peak Stress (MPa) | | | Peak Absorption Energy (J) | | |
|---|---|---|---|---|---|---|
| | 20%-3-90° | 20%-4-90° | 20%-5-90° | 20%-3-90° | 20%-4-90° | 20%-5-90° |
| 3800 | 72.4 | 81.7 | 111 | 47.3 | 275.6 | 301.1 |
| 4100 | 204.8 | 211 | 227 | 282.8 | 413.4 | 620.1 |
| 5100 | 236.5 | 255.9 | 272 | 376.4 | 475.6 | 665.3 |
| 6100 | 303.9 | 304 | 307 | 537.7 | 597.7 | 762.8 |

Note: the number 20% denotes the mass fraction of STF; the numbers 3, 4, and 5 all represent the number of layers; and the number 90° represents the layer angle.

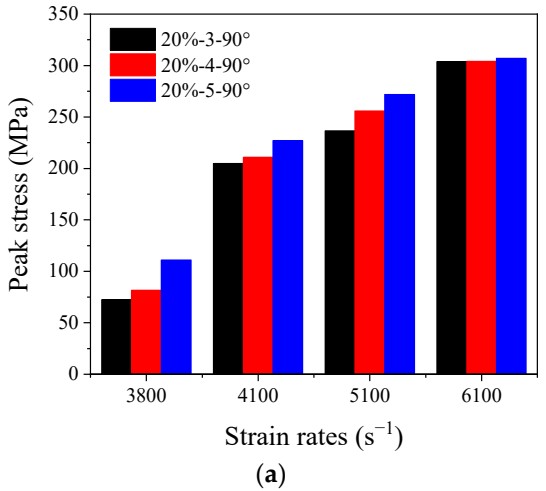
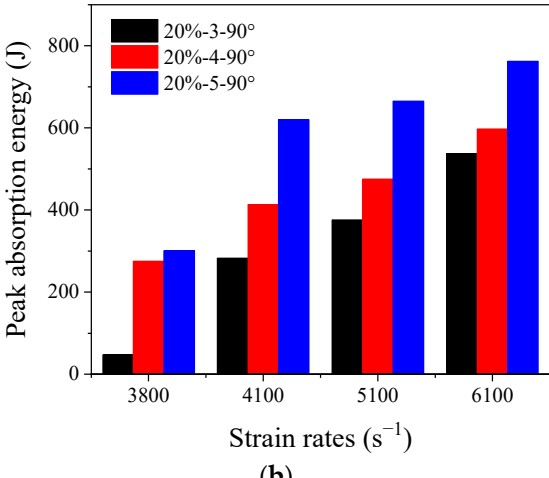

(**a**)　　　　　　　　　　　　　(**b**)

**Figure 9.** Effect of the layer number on the impact response for GFRP-STF specimen under different strain rates. (**a**) Peak stress; (**b**) peak absorption energy.

## 4. Conclusions

This study involved conducting impact tests using a SHPB on four neat GFRP and twenty GFRP-STF specimens at high strain rates. Both neat GFRP and GFRP-STF samples displayed substantial strain-rate effects, but the GFRP-STF specimens demonstrated more notable effects than the GFRP samples. This distinction is reflected in the graphs of stress–strain and energy-absorption time, as well as peak stress and energy-absorption results. STF immersion significantly amplified the peak stress of GFRP under high strain rates, with the degree of enhancement increasing with the strain rate. Crucially, STF can alter GFRP's mechanical response at high strain rates, leading to a noteworthy yield stage, which becomes more prominent with an increasing strain rate. GFRP impregnation with STF also leads to a considerable surge in energy absorption capacity under high strain rates, though the magnitude of improvement is not monotonous.

The laying angle of GFRP-STF composites exerts a notable effect on their high-strain-rate response, though the pattern is not uniform. Notably, a laying angle of 45° results in the most significant increase (up to 130.2%) in peak stress. However, while GFRP-STF composites with a laying angle of 90° have an optimal advantage in terms of energy-absorption capacity, this advantage is insignificant compared to the composite specimens with a laying angle of 45°. Increasing the number of layers heightens both peak stress and energy absorption, although the effect of laying layers on peak stress increase is minimal. Furthermore, due to the better deformation ability of GFRP-STF composites compared to GFRP, laying layers exerts a notable impact on energy-absorption capacity, particularly when impact energy is substantial.

**Author Contributions:** For research articles with several authors, the following statements should be used Conceptualization, L.S. and M.W.; methodology, L.S.; software, W.G.; validation, L.S., M.W. and W.G.; formal analysis, M.W.; investigation, M.W.; resources, L.S.; data curation, W.G.; writing—original draft preparation, M.W.; writing—review and editing, M.W.; visualization, L.S.; supervision, L.S.; project administration, L.S.; funding acquisition, L.S. All authors have read and agreed to the published version of the manuscript.

**Funding:** This research received no external funding.

**Institutional Review Board Statement:** Not applicable.

**Informed Consent Statement:** Not applicable.

**Data Availability Statement:** The data that support the findings of this study are available from the corresponding author upon request.

**Acknowledgments:** The authors acknowledge financial support from the National Natural Science Foundation of China (no.: 52078310), Liaoning Revitalization Talents Program (no.: XLYC1902038), and Natural Science Foundation of Zhejiang Province (grant number LY23E080011). The authors are extremely grateful to the anonymous reviewers for their valuable criticisms and useful suggestions that aided in improving the quality of the present study as well as future work.

**Conflicts of Interest:** The authors declare that they have no known competing financial interest or personal relationships that could have appeared to influence the work reported in this paper.

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
