# Peer review of "High Strain Rates Impact Performance of Glass Fiber-Reinforced Polymer Impregnated with Shear-Thickening Fluid"

_jcs, doi:10.3390/jcs7050208_

Round 1

Reviewer 1 Report (Previous Reviewer 2)

The authors did some work to explain the results. Indeed, it showed that the work is voluminous and extensive and includes a series of publications.

The authors indicated the reason for the manifestation of the effect of the mechanical stability of GFRP-STF under high strain rate impact. And they highlighted the novelty of the manuscript in question.

However, the work has not been significantly altered, the data presented are not significant.

I would like to see a more exhaustive study of the material under consideration to results been published.

The language of presentation is clear and accessible.

Author Response

Dear reviewer,

Thank you for your thoughtful comments and suggestion.

Our team has researched the impact mechanical behavior of STF and FRP-STF composite materials. In terms of low-velocity impact with a drop hammer, we conducted a detailed study on the influence of physical parameters of GFRP itself on its low-velocity impact mechanical behavior, such as "Effects of parameters controlling the impact resistance behavior of the GFRP fabric embedded with a shear thickening fluid," published in "Materials and Design"; The effect of STF on the mechanical behavior of different types of FRP under the low-velocity impact has also been investigated, such as "Low-velocity impact the performance of FRP embedded with shear thickening fluid," published in "Polymer Testing." In terms of high strain rate impact, we conducted a study on the strain rate effect of multi-walled carbon nanotube (MWCNT) reinforced STF under high strain rate impact, such as "High strain rate characterization of shear thickening fluid reinforced with MWCNT nanoparticles using SHPB," published in "Materials Letters"; At the same time, we also conducted a study on the mechanical behavior of GFRP-MWCNT/STF composite materials under high strain rate impact (DOI: 10.1007/s10443-023-10131-x), which has been accepted by "Applied Composite Materials." In addition, we have conducted detailed research on AFRP, BFRP, and CFRP, and the relevant results are currently being compiled. We look forward to receiving your professional comments again when these works are published. The comments help us to improve the reliability and systematicity of our work.

On the other hand, based on your suggestion, we will conduct a study on the impact mechanical behavior of GFRP-STF under the influence of environmental temperature. Our research goal is to achieve the application of GFRP-STF in engineering environments.

We sincerely appreciate the reviewers’ warm work, hopefully we have addressed all your concerns.

Yours sincerely

Dr. Minghai Wei

Reviewer 2 Report (Previous Reviewer 4)

Accept after minor revision (corrections to minor methodological errors and text editing)

Minor editing of English language required

Author Response

Dear reviewer,

Thank you for your careful review.

The Manuscript-R1 has been proofread by a native English speaker.

Yours sincerely

Dr. Minghai Wei

Round 2

Reviewer 2 Report (Previous Reviewer 4)

Accept in present form

This manuscript is a resubmission of an earlier submission. The following is a list of the peer review reports and author responses from that submission.

Round 1

Reviewer 1 Report

The work looks scientifically sound. It’s difficult for me to judge about the novelty and significance of the content. I suggest acceptance after minor revisions. The following revisions should be considered.

1. In materials section. GFRP – the authors should describe what specific polymer is hidden under “P”. It will be easier for the reader to compare the data with other articles then. Different polymers have different features. Please explain what is the polymer.

2. The SEM technique should be appropriately described. I wonder, how did the authors made the SEM pictures, given. That PEG-200 is volatile under vacuum. What regimes did they use. I suppose that there are no high-resolution pictures because PEG-200 is volatile.

3. Because there are no high resolution images, where the nanoparticles are visible (the highest magnitude is 5K, 10 mkm) each time the authors write about nanoparticles and refer to SEM images is an assumption. There are several issues in the text, but for instance: “ As shown in Fig. 3, this can be attributed to the GFRP impregnated with the STF containing numerous nanoparticles filling and covering the fibers”. I don’t understand what is “shown on Fig. 3” that gives the authors right to imply that “ numerous nanoparticles filling and covering the fibers”.

4. In the text Fig. 10 is mentioned after the fig. 6, and before the Fig. 7. Generally, the figure should follow after it was first mentioned in the text.

Reviewer 2 Report

The manuscript was created by a team of authors who have considerable experience in the field of modifying woven materials to give specific mechanical properties. The article appears to be well prepared with a fair amount of research done. The manuscript is of interest to a small group of researchers and practitioners, because reflects a very small part of the materials studied by the author and there is no novelty in the presented materials.

In addition, the conclusions presented about the effective number of layers of the modifying material, as well as the optimal angle for applying the modifying material, have already been formulated 10.1016/j.matdes.2020.109078

1. In addition, in sections 2.1-2.2, the authors need to expand the description of the technique for applying modifying layers and show how different deposition angles can be provided.

2. It would be important to provide generalized data (table) on mechanical parameters for all modified samples (at different application angles, different number of layers).

3. In section 3 please describe the reasons for the change in the mechanical behavior of the modified materials. (relative to Absorption energy-time of neat GFRP and GFRP-STF specimens)

Reviewer 3 Report

Dear authors, 

I cannot accept the article for publication as it contains substantial and methodological errors.

Figure 4 shows that the specimen is larger than the bars, which is inconsistent with the SHPB test methodology.

Figure 6 shows that the absorbed energy is decreasing, which is inconsistent with the physics of the phenomenon. Besides, the velocity of sound in the aluminium used is 5082.3 m/s, which, with a striker length of 300 mm, should give an impulse duration of 118 ms. The figure shows that the impulse lasts about 150 ms.

Determining Young's modulus using the SHPB method is not trivial. Among other things, it is related to the achievement of dynamic equilibrium in the system. Therefore, it is necessary to provide information on when the system reached equilibrium and how the parameter R describing the dynamic equilibrium in the system changed.

Section 2.4 should be modified in analogy with other publications in this field. Among other things, the equations used to obtain the stress-strain curves are absent, as is the method of determining the energy absorbed by the specimen.

The authors write that the transmitted signal was weak, then how was it amplified? What was the bandwidth of the amplifier? Does this affect the results of the study?

There is no information about the pulse shaper, strain gauges and data acquisition system.

 Below are some of the literature items to which the authors should refer:

Song, B.; Chen, W., Dynamic stress equilibration in split Hopkinson pressure bar tests on soft materials, Experimental Mechanics, 10.1007/BF02427897

Chen, W.; Song, B., Split Hopkinson (Kolsky) bar: design, testing and applications, 2011

W. Zhong, A. Rusinek, Influence of interfacial friction and specimen configuration in Split Hopkinson Pressure Bar system , DOI:10.1016/J.TRIBOINT.2015.04.002

R. Panowicz, J. Janiszewski, K. Kochanowski, Numerical and Experimental Studies of a Conical Striker Application for the Achievement of a True and Nominal Constant Strain Rate in SHPB Tests, Experimental Mechanics

Reviewer 4 Report

The study examined how a glass fiber-reinforced polymer fabric with a shear-thickening fluid performs under high-strain-rate conditions using a split Hopkinson pressure bar. The results showed that the composite material could withstand higher and faster impact loads and had better deformation ability and toughness compared to the plain GFRP. The optimal laying angle was found to be 45° for peak stress and 90° for energy absorption capacity, while the number of layers had a significant impact on energy absorption.

1.  In order to better highlight the advantages of this work, the author needs to provide a table to compare related work. 

2. A legible scale bar should be added to figures 1, 3, 8 .

3. What is the morphology of GFRP and GFRP-STF specimens after test in this work? The author needs to give some proof.